# Evaluation of the Anticancer Activity and Mechanism Studies of Glycyrrhetic Acid Derivatives toward HeLa Cells

**DOI:** 10.3390/molecules28073164

**Published:** 2023-04-02

**Authors:** Ju Chen, Yunran Xu, Yan Yang, Xin Yao, Yuan Fu, Yi Wang, Yunjun Liu, Xiuzhen Wang

**Affiliations:** 1School of Pharmacy, Guangdong Pharmaceutical University, Guangzhou 510006, Chinalyjche@gdpu.edu.cn (Y.L.); 2Department of Pharmacy, Guangdong Second Provincial General Hospital, Guangzhou 510317, China; 3Guangdong Provincial Key Laboratory of Advanced Drug Delivery, Guangdong Provincial Engineering Center of Topical Precise Drug Delivery System, Guangdong Pharmaceutical University, Guangzhou 510006, China

**Keywords:** glycyrrhetinic acid derivatives, esterification, antitumor, apoptosis, RNA-sequence

## Abstract

In this paper, a series of glycyrrhetic acid derivatives **3a**–**3f** were synthesized via the esterification reaction. The cytotoxicity of these compounds against five tumor cells (SGC-7901, BEL-7402, A549, HeLa and B16) and normal LO2 cells was investigated using the 3-(4,5-dimethylthiazol-2-yl)-2,5-diphenyltetrazolium bromide (MTT) method. The results showed that compound **3a** exhibited high antiproliferative activity against HeLa cells (IC_50_ = 11.4 ± 0.2 μM). The anticancer activity was studied through apoptosis, cloning, and scratching; the levels of the intracellular ROS, GSH, and Ca^2+^; and the change in the mitochondrial membrane potential, cell cycle arrest and RNA sequencing. Furthermore, the effects of compound **3a** on gene expression levels and metabolic pathways in HeLa cells were investigated via transcriptomics. The experimental results showed that this compound can block the cell cycle in the S phase and inhibit cell migration by downregulating Focal adhesion kinase (FAK) expression. Moreover, the compound can reduce the intracellular glutathione (GSH) content, increase the Ca^2+^ level and the intracellular ROS content, and induce a decrease in the mitochondrial membrane potential, further leading to cell death. In addition, it was also found that the mechanism of compounds inducing apoptosis was related to the regulation of the expression of mitochondria-related proteins B-cell lymphoma-2 (Bcl-2), Bcl-2-Associated X (Bax), and the activation of the caspase proteins. Taken together, this work provides a help for the development of glycyrrhetinic acid compounds as potential anticancer molecules.

## 1. Introduction

Pentacyclic triterpenoids are an important class of natural product, and these compounds have a variety of biological activities, thus attracting the in-depth study of scientists [1]. 18β-glycyrrhetinic acid (18β-GA) is a glycyrrhizin aglycone found in the licorice root, which has anti-inflammatory, antiulcer, and antibacterial activities. In addition, 18β-GA has the advantages of easy availability, low cost, good stability, and high biosafety [2,3]. However, the disadvantage of glycyrrhetinic acid (GA) is its weak ability to inhibit tumor cell proliferation [4,5]. To enhance the cytotoxicity of GA, many researchers have synthesized a number of GA derivatives [6,7,8,9,10]. For example, Rui et al. found that the C-30 position of 18β-GA modified with an electron-withdrawing group can improve its toxicity toward tumor cells [11]. Moreover, Csuk et al. performed esterification and amidation reactions on C-30 of GA and the synthesized compounds (esters derivatives) increase cytotoxic activity [12]. These compounds can block the cell cycle and induce apoptosis by modulating the expression of signaling proteins and mitochondrial function [13,14]. The studies on the cytotoxicity show that GA derivatives are potential and promising antitumor drugs [15,16,17,18]. In this study, we used the esterification of GA with different substituent phenol at the para position to synthesize a series of GA derivatives **3a**–**3f** (Figure 1). Moreover, the synthetic compounds were characterized using HRMS, NMR spectra. To investigate the anticancer activity of compounds **3a**–**3f**, we used the 3-(4,5-dimethylthiazol-2-yl)-2,5-diphenyltetrazolium bromide (MTT) method to evaluate the cytotoxicity of compounds against several cancer and normal cells. We also studied the apoptotic efficacy of the compounds on HeLa cells and used western blot analysis to detect the expression levels of some apoptosis-related proteins. This study shows that the compounds induce apoptosis through the mitochondrial pathway. Additionally, the effect of the compounds on intracellular gene expression was studied via transcriptomics, and it was discovered that the compounds could affect the metabolic pathways of cells and cause the decrease in the expression of GPX4, which is an important regulator of cellular iron death.

## 2. Results and Discussion

### 2.1. Chemistry

The synthetic route of glycyrrhetinic acid derivatives **3a**–**3f** is shown in Figure 1. Through 1-ethyl-3-(3-dimethylaminopropyl) carbodiimide hydrochloride (EDCI) and 4-dimethylaminopyridine (DMAP)-mediated acylation, the carboxyl group of GA was activated to form amide and ester bonds. Under the catalysis of EDCI and DMAP, 18β-glycyrrhetic acid underwent esterification with p-bromophenol to produce compound **3a** at room temperature and was purified via column chromatography using a mixture of EtOAc-petroleum ether (1:3, *v*/*v*) as the eluent. The purity of compound **3a** was detected via HPLC (COSMOSIL Packed Column 5C18-MS-II 20ID × 250 mm, Model: K1504557) using a mixture of methanol and water (*v*:*v* = 95:5) as the mobile phase. In Appendix A, only one peak was observed within 30 min, which indicates that compound **3a** is pure and the purity is 99.51%. The melting point of compound **3a** was also found to be 205.5–205.9 °C via the capillary method. The molecular weights of these compounds detected in the HRMS spectra were consistent with the expected values. Moreover, the structures of the compounds were also examined using ^13^C NMR spectra and ^1^H NMR spectra. In the ^1^H NMR spectra, the peaks at 3.01, 4.30–4.31, and 5.42–5.45 ppm are assigned to the hydrogen atoms of H_a_, H_b,_ and H_c_, respectively. In the ^13^C NMR spectra, the chemical shifts 199.5, 175.1, 169.5 ppm for **3a**, 199.5, 175.1, 169.5 ppm for **3b**, 199.4, 175.3, 169.5 ppm for **3c**, 199.5, 174.7, 169.4 ppm for **3d**, 199.5, 175.3, 169.6 ppm for **3e**, and 199.5, 175.5, 169.6 ppm for **3f** are assigned to the C (11), C (30), C (13), respectively, while the peaks of 77.0 for **3a**–**3f** are attributed to C (3). 

### 2.2. Cell Viability and IC_50_ Determination

According to previous reports, the introduction of the ester group at the C-30 site can enhance the biological activity of GA [19,20]. To determine the antitumor activity of compounds **3a**–**3f**, the MTT assay was used to evaluate the anti-proliferative effects of the compounds against five tumor cells and normal LO2 cells. The IC_50_ values of compounds are listed in Table 1. The anti-proliferative effects of **3a**–**3d** on selected five tumor cells was higher than that of GA (except **3c** against BEL-7402). In particular, the IC_50_ data of compound **3a** (IC_50_ = 11.4 ± 0.2 μM) against HeLa cells were about 6 times greater than that of GA. We also found that the GA derivatives with the electron-withdrawing group such as -X or NO_2_ possessed a higher ability to inhibit the proliferation of the five types of tumor cells than those with electron-donating groups (such as **3e** and **3f**). Zheng et al. reported that the benzyl introduced at the C-30 position of GA had an IC_50_ value of 17.53 μM against HeLa [21]; the introduction of halogen atoms into the compounds showed better antitumor activity [22]. Therefore, the electron-withdrawing group can enhance the activity of the GA derivatives. The cell viability of **3a** toward HeLa cells is depicted in Figure 1; the cell viability decreases with increasing concentrations of **3a**. Because compound **3a** shows the highest ability to suppress HeLa cell growth, we chose compound **3a** to study its antitumor mechanism on HeLa cells.

### 2.3. Apoptosis Studies

To determine whether the compound inhibiting the cell proliferation is attributed to apoptosis in HeLa cells, 4′,6-diamidino-2′-phenylindole (DAPI) staining was used to analyze the cell morphology. The cells were treated with 10 μM, 20 μM, and 30 μM of compound **3a** and stained with DAPI, as shown in Figure 2a; the cell morphology in the control group was intact. However, the cells treated with compound **3a**, i.e., the apoptotic cells showed apoptotic features such as nuclear shrinkage and chromatin condensation. The morphologic changes of the nuclear cells confirmed that the compound **3a** induced the apoptosis. To further evaluate the ability of compound **3a** to induce apoptotic cell death, HeLa cells were exposed to compound **3a** and the apoptosis was quantitatively analyzed using flow cytometry. As shown in Figure 2b, the percentage of apoptotic cells in the control group is 3.07%, while the apoptotic percentage increased by 4.50% for 10 μM, 4.56% for 20 μM, and 14.23% for 30 μM of **3a** compared with that of the control. The results indicate that the compound **3a** can effectively induce apoptosis in a dose-dependent manner.

### 2.4. Inhibition of Cell Migration and Cloning

Cell migration is involved in many biological processes that are closely related to tumor development and metastasis of cancer cells [23,24]. Therefore, the inhibition of cancer cell migration is crucial for the compound exerting anticancer activity. We used scratch experiments to study the effect of compound **3a** on cell migration. After treatment of HeLa cells with different concentrations of compound **3a** for 24 h, the migration of cells was observed. As shown in Figure 3a, the width in the control group decreased obviously; however, the distance of wounds edge of **3a**-treated groups showed less changes compared with the control group. As shown in Figure 3b, the quantification analysis also showed that the percent of wound closure in the different concentration of compound **3a**-treated groups were different; 30 μM of **3a** showed the highest inhibitory effect on the cell migration. Focal adhesion kinase (FAK) protein plays an important role in the process of cell proliferation [25]; when its expression is blocked, cell invasion and metastasis are inhibited [26]. As shown in Figure 3c,d, the expression level of FAK decreased significantly with the increasing concentration of compound **3a**, which further indicated that the compound **3a** can effectively inhibit the metastasis of cancer cells. Additionally, the cell cloning was studied; as shown in Figure 3e, after the cells were treated with compound **3a** for 10 continuous days, the number of the living cells decreased compared with that in the control, indicating that **3a** can prevent the cell colonies formation. The results demonstrated that compound **3a** can inhibit the cell migration and colony formation in a dose-dependent manner.

### 2.5. Cell Cycle Arrest Assay

To evaluate the effect of the compound **3a** on cell cycle distribution, HeLa cells were exposed to different concentrations of compound **3a** for 24 h and analyzed via flow cytometry. As shown in Figure 4a, compared with the control group, we found that the cell proportion in the S phase increased by 4.56% for 10 µM, 9.65% for 20 µM, and 20.81% for 30 µM, accompanied by a reduction in the G0/G1 phase, which indicated that the compound **3a** inhibited the cell proliferation in the S phase in a concentration-dependent manner. In addition, we used western blot to detect the expression level of cell cycle arrest-related proteins (p53, p21), and the results are shown in Figure 4b. Through the quantitative analysis of p53 and p21, the gray values of p53 and p21 proteins showed an upward trend compared with that in the control (Figure 4c), which further verified that the compound **3a** blocked the cell cycle in the S phase.

### 2.6. Determination of Intracellular Reactive Oxygen Species (ROS)

Recent studies have shown that mitochondria are the main source of ROS production in mammalian cells [27]; excessive accumulation of ROS can lead to mitochondrial damage and cell death [28,29]. To demonstrate whether compound **3a** can increase intracellular ROS content, we used 2′,7′-dichlorofluorescein diacetate (DCFH-DA) as a fluorescent indicator to analyze the intracellular ROS levels, and Rosup was used as a positive control. As shown in Figure 5a, in the control group, only a weak green fluorescence was detected, while the green fluorescence was significantly enhanced after the cells were treated with different concentration of compound **3a**, indicating that compound **3a** caused an increase of intracellular ROS levels. Additionally, the quantitative determination of the intracellular ROS levels was performed using flow cytometry (as shown in Figure 5b). Compared with the control group, the DCF fluorescence intensity in the cells increased in a concentration-dependent manner. At low concentration (10 μM), the increase in fluorescence intensity was not obvious, and when the concentration of **3a** was 20 μM and 30 μM, the fluorescence intensity of DCF increased to 1.4-fold and 1.8-fold, respectively. These results suggest that compound **3a** increases the generation and accumulation of intracellular ROS and triggers apoptosis by disrupting ROS homeostasis.

Excessive ROS production also leads to oxidative stress and activates the mitogen-activated protein kinase (MAPK) signaling pathway [30]. In response to stress, p38MAPK can be activated and phosphorylated [31]. The activation of p38MAPK induces cancer cell apoptosis and inhibits tumor formation [32]. As shown in Figure 5c, we detected the expression of p38MAPK protein using western blot and found that the expression of p38MAPK was upregulated after HeLa cells were treated with compound **3a**. The above results suggest that compound **3a** causes excess intracellular ROS to lead to oxidative stress and ultimately induce apoptosis, which is associated with the activation of the p38MAPK pathway.

### 2.7. Effects of Compound on Ca^2+^ Levels

As a second messenger of various death signal transduction, Ca^2+^ has an intricate relationship with apoptosis. Elevated levels of intracellular Ca^2+^ disrupt the mitochondrial membrane potential, and further disrupt the electron transport chain, generate excessive ROS, and eventually lead to apoptosis [33,34]. A Fluo-3AM fluorescent probe was used to examine the effect of compound **3a** on intracellular Ca^2+^ levels. As shown in Figure 6a, the green fluorescence in the control group was very weak, while the bright green fluorescence spots gradually increased after treatment of HeLa cells with different concentrations of compound **3a**. As shown in Figure 6b, the intracellular Ca^2+^ content was significantly higher in the compound **3a**-treated group than in the control group. This indicates that compound **3a** can increase the level of intracellular Ca^2+^; therefore, the balance of intracellular Ca^2+^ is damaged. Excessive intracellular Ca^2+^ concentration further disrupts mitochondrial function, resulting in the release of cytochromes, which prevents cancer cells from growing.

### 2.8. Effects on Mitochondrial Membrane Potential

The mitochondria play an important role in both extrinsic and intrinsic apoptosis, and a reduction in the mitochondrial membrane potential (MMP) is considered an early event in apoptotic cells [35,36]. To further investigate whether apoptosis of HeLa cells is related to mitochondrial dysfunction, we determined the change of the mitochondrial membrane potential with 5,5′,6,6′-Tetrachloro-1,1′,3,3′-tetraethyl-imidacarbocyanine iodide (JC-1) as a fluorescent probe. At high MMP, JC-1 emits red fluorescence; at low MMP, JC-1 emits green fluorescence. As shown in Figure 7a, compared with the control, MMP was significantly decreased in HeLa cells treated with cccp (positive control) and different concentrations of compound **3a** (10, 20 and 30 μM), manifested by a gradual increase in JC-1 monomer (appeared as green fluorescence, indicating mitochondrial depolarization), while the JC-1 polymer (appeared as red fluorescence, indicating mitochondrial hyperpolarization) gradually decreased. Therefore, compound **3a** can induce a decrease in the mitochondrial membrane potential. To further quantify the changes in the mitochondrial membrane potential, we used flow cytometry to measure the ratio of red/green fluorescence intensities. As shown in Figure 7b, the red/green fluorescence ratio in the control group is 2.56; with increasing concentration of compound **3a**, the red/green fluorescence ratio gradually decreased. All the above results indicate that compound **3a** can significantly induce the decrease in the MMP.

### 2.9. Analysis of Intracellular GSH Content

It is widely believed that the imbalance between cellular antioxidant capacity and reactive oxygen species (ROS) formation induced by intracellular glutathione (GSH) leads to the death of tumor cells [37,38]. GSH can bind to excess oxygen radicals, and the decrease in GSH levels can lead to increased oxygen radicals in the cell, which can affect normal cell growth. [39,40]. As shown in Figure 8a, the intracellular GSH content in the control is 57.7; after HeLa cells were treated with compound **3a** for 24 h, a decrease in the intracellular GSH content of 10.4% for 10 μM, 18.2% for 20 μM, and 24.3% for 30 μM was observed. In Figure 8b, the ratio of GSH/GSSG (glutathione disulfide) was determined. In the control, the ratio of GSH/GSSG is 9.9; after an exposure of HeLa cells to 10, 20, and 30 μM of compound **3a** for 24 h, the ratios of GSH/GSSG reduced compared with that in the control. Therefore, compound **3a** can reduce the content of intracellular GSH and cause an increase in oxidant stress.

### 2.10. Differential Gene Expression Level Analysis

To investigate the antitumor mechanism of compound **3a**, the effect on transcriptional genomic expression of HeLa cells was assayed through high-throughput sequencing. In the differential gene expression statistics and volcano (Figure 9a,b), compound **3a** could affect the expression of 50 genes, in which the expression of 31 genes was downregulated and 19 genes were upregulated. KEGG metabolic pathway analysis (Figure 9c) showed that these differential genes were mainly enriched in phagosomes, iron death, primary immunodeficiency, and some biometabolic processes. The GO enrichment analysis showed that the biological processes were regulated by these differential genes as well as their molecular functions (Figure 9d). This further demonstrated that compound **3a** affected the activity of some intracellular nucleotide receptors and intracellular signaling including G protein-coupled receptor signaling pathway, intracellular production of some factors and other biological processes. In addition, the above results suggest that the anticancer activity may be related to processes including cytokine secretion, glycoconjugate metabolism, signaling, and iron death of cells.

### 2.11. Western Blot Detection

To further explore the molecular mechanism of apoptosis, the effect of compound **3a** on the apoptosis-related protein was investigated using western blot. According to literature, the BH3 family of proteins plays a crucial role in the regulation of intrinsic apoptotic pathways [41]. After the cells are stimulated by apoptotic factors, the various apoptotic signals can be activated and the change in the expression of corresponding proteins, including pro-apoptotic proteins (Bax, Bad) and anti-apoptotic proteins (Bcl-2), is observed. The expression of Bax protein induces the opening of transport pores in the mitochondrial membrane to release cytochrome C, which binds to apoptotic protein activating factor (Apaf-1) and activates caspase-9. The activated caspase-9 further activates caspase-7 and caspase-3. The activated caspase-3 can cleave the cellular substrate PARP, eventually leading to apoptosis [42]. As shown in Figure 10a,b, we found that compound **3a** significantly upregulated the expression of the pro-apoptotic protein Bax, while downregulating the expression of the anti-apoptotic protein Bcl-2 compared with that in the control. Furthermore, after HeLa cells were treated with compound **3a** for 24 h, the apoptosis-executing protein caspase-3 was activated, resulting in upregulation of its downstream molecule caspase-7 and increased cleavage of PARP. RNA sequencing experiments show that compound **3a** can also induce iron death in HeLa cells. It is well known that the marker of iron death is the inactivation of GPX4 [43], so we also detected the effect of compound **3a** on the expression level of GPX4 using protein blotting experiments. As observed in Figure 10, compound **3a** downregulated the expression of GPX4 protein, indicating that compound **3a** caused iron death in tumor cells; this is consistent with the results of KEGG metabolic pathway analysis.

## 3. Materials and Methods

### 3.1. Materials

A Bruker AVANCE-500 spectrometer was used to detect NMR spectra. All chemical shifts were given relative to tetramethylsilane (TMS). Bruker 7.0 T SolariX XR FT-ICR-MS was used to record mass spectra. TLC-analysis was performed on glass-backed plates (Sigma-Aldrich, Canada) coated with 0.2 mm silica 60F254. Commercial common reagent-grade chemicals were used without further purification. The gastric adenocarcinoma cell line SGC-7901, cervical cancer cell line HeLa, lung carcinoma cell line A549, human hepatocellular carcinoma cell line BEL-7402, and normal live cell line LO2 were purchased from the cell bank of the Cell Institute of Sinica Academia Shanghai (Shanghai, China). Buffers were prepared using doubly distilled water. The 4′,6-diamidino-2′-phenylindole (DAPI), cell cycle and apoptosis analysis kits were purchased from Beyotime (Shanghai, China). 3-(4,5-dimethylthiazol-2-yl)-2,5-diphenyl-2H-tetrazolium bromide (MTT) was obtained from Sigma–Aldrich. The fluorescent dye 2′,7′-dichlorodihydrofluorescein diacetate (DCHF-DA) and 5,5′,6,6′-tetrachloro-1,1′,3,3′-tetraethylbenzimidazolcarbocyanine iodide (JC-1) were purchased from Roche Diagnostics (Indianapolis, IN, USA). Polyclonal antibodies against Bcl-2, Bax, and P38 were purchased from Santa Cruz Biotechnology (Santa Cruz, CA, USA). Caspase-3 antibodies were purchased from Cell Signaling Technology (Beverly, MA, USA).

### 3.2. Synthesis of Compounds

A mixture of 18β-glycyrrhetinic acid (18-GA) (0.471 g, 1 mmol), 4-dimethylaminopyridine (DMPA) (0.214 g, 1 mmol), and 1-ethyl-3-(3-dimethylaminopropyl) carbodiimide (EDCI) (0.283 g, 1.5 mmol) in dichloromethane (DCM) was stirred for 30 min; then, substituent phenol (1 mmol) was added and stirred for 12 h at room temperature. Thin layer chromatography (TCL) was used to monitor the reaction. The mixture was extracted in DCM. The organic phase was dried over using MgSO_4_. The solvent was removed, and the crude product was purified via column chromatography on silica gel (100–200 mesh) with a mixture of EtOAc-petroleum ether (1:3, V/V) as the eluent; a white power was obtained.

4-Bromophenyl 3β-hydroxy-11-oxo-olean-12-en-30-oate (**3a**): Yield 61.2%, ^1^H-NMR (500 MHz, DMSO-*d*_6_) (Appendix A): δ 7.62 (d, *J* = 8.5 Hz, 2H, Ar-H), 7.10 (d, *J* = 8.5 Hz, 2H, Ar-H), 5.43 (s, 1H), 4.31 (s, 1H), 3.01 (t, 1H, *J* = 6.0 Hz, 1H), 2.64–2.50 (m, 4H), 2.36–2.33 (m, 2H), 2.13–2.11 (m, 3H), 1.95–1.77 (m, 6H), 1.65–1.63 (m, 2H), 1.59–1.50 (m, 4H), 1.19–1.16 (m, 2H), 1.10–1.05 (m, 6H), 1.02–0.91 (m, 6H), 0.90–0.82 (m, 3H), 0.81–0.68 (m, 4H). ^13^C-NMR (125 MHz, DMSO-*d*_6_) (Appendix A): 199.5, 175.1, 169.5, 150.2, 132.9 (2C), 127.9, 124.5(2C), 118.6, 77.0, 61.6, 54.5, 48.6, 45.3, 44.3, 43.3, 40.1, 39.9, 39.8, 37.7, 37.1, 32.5, 32.1, 30.6, 28.6, 28.5, 27.8, 27.4, 26.5, 26.1, 23.4, 18.8, 18.4, 17.9, 17.6. HRMS (solvent) calcd for C_36_H_49_BrNaO_4_^+^: *m*/*z* = 647.2712 ([M + Na]^+^), found: *m*/*z* = 647.2517 (Appendix A).

4-Chlorophenyl 3β-hydroxy-11-oxo-olean-12-en-30-oate (**3b**): Yield 65.3%, ^1^H-NMR (500 MHz, DMSO-*d*_6_) (Appendix A): δ 7.50 (d, *J* = 8.5 Hz, 2H, Ar-H), 7.19–7.15 (m, 2H, Ar-H), 5.43 (s, 1H), 4.31 (s, 1H), 3.01 (t, 1H, *J* = 6.0 Hz, 1H), 2.63 (s, 1H), 2.58–2.55 (m, 2H), 2.37–2.33 (m, 2H), 2.19–2.12 (m, 3H), 1.97–1.73 (m, 6H), 1.69–1.62 (m, 1H), 1.59–1.50 (m, 4H), 1.48–1.42 (m, 2H), 1.39–1.31 (m, 4H), 1.18–1.14 (m, 2H), 1.11–0.92 (m, 6H), 0.91–0.81 (m, 4H), 0.80–0.69 (m, 5H). ^13^C-NMR (125 MHz, DMSO-*d*_6_) (Appendix A): 199.5, 175.1, 169.5, 156.7, 149.7, 130.4, 130.0, 129.5, 127.9, 124.1, 122.7, 117.3, 77.0, 61.6, 54.5, 48.5, 45.3, 44.3, 43.3, 40.1, 39.9, 39.8, 37.7, 37.1, 32.5, 32.1, 30.6, 28.6, 28.5, 27.8, 27.4, 26.5, 26.1, 23.4, 18.7, 17.6. HRMS (CH_3_OH) calcd for C_36_H_49_ClNaO_4_^+^: *m*/*z* = 603.3217 ([M + Na]^+^), found: *m*/*z* = 603.2985 (Appendix A).

4-Fluorophenyl 3β-hydroxy-11-oxo-olean-12-en-30-oate (**3c**): Yield 65.3%, ^1^H-NMR (500 MHz, DMSO-*d*_6_) (Appendix A): δ 7.26 (d, *J* = 8.5 Hz, 2H), 7.15 (t, *J* = 4.5 Hz, 2H), 5.43 (s, 1H), 4.31 (s, 1H), 3.01 (t, 1H, *J* = 6.0 Hz, 1H), 2.59–2.48 (m, 3H), 2.37–2.33 (m, 2H), 2.19–2.11 (m, 2H), 1.98–1.73 (m, 4H), 1.69–1.62 (m, 1H), 1.60–1.51 (m, 4H), 1.46–1.32 (m, 5H), 1.18–1.14 (m, 1H), 1.12–1.02 (m, 8H), 0.97–0.92 (m, 4H), 0.89–0.81 (m, 4H), 0.78–0.71 (m, 4H). ^13^C-NMR (125 MHz, DMSO-*d*_6_) (Appendix A): 199.4, 175.3, 169.5, 161.0, 159.1, 147.0, 127.9, 124.0, 123.9, 116.7, 116.5, 77.0, 61.6, 54.5, 48.5, 45.3, 44.3, 40.1, 39.9, 39.8, 37.1, 32.5, 32.1, 30.6, 28.6, 28.5, 27.8, 27.4, 26.9, 26.5, 26.1, 23.4, 18.7, 17.6, 16.5, 16.4. HRMS (CH_3_OH) calcd for C_36_H_49_FNaO_4_^+^: *m*/*z* = 587.3513 ([M + Na]^+^), found: *m*/*z* = 587.3310 (Appendix A).

4-Nitrophenyl 3β-hydroxy-11-oxo-olean-12-en-30-oate (**3d**): Yield 65.3%, ^1^H-NMR (500 MHz, DMSO-*d*_6_) (Appendix A): δ 8.32 (d, *J* = 9.0 Hz, 2H, Ar-H), 8.11 (d, *J* = 9.0 Hz, 2H, Ar-H), 5.45 (s, 1H), 4.31 (s, 1H), 3.01 (t, 1H, *J* = 6.0 Hz, 1H), 2.58–2.55 (m, 1H), 2.38–2.34 (m, 2H), 2.21–2.13 (m, 2H), 1.96–1.77 (m, 4H), 1.67–1.44 (m, 4H), 1.42–1.35 (m, 16H), 1.19–1.14 (m, 1H), 1.10–1.02 (m, 6H), 0.96–0.88 (m, 2H), 0.87–0.79 (m, 2H), 0.78–0.65 (m, 2H). ^13^C-NMR (125 MHz, DMSO-*d*_6_) (Appendix A): 199.5, 174.7, 169.4, 164.4, 155.9, 145.6, 128.0, 126.6, 125.8 (2C), 123.6 (2C), 116.2, 77.0, 61.6, 54.5, 48.5, 45.3, 44.5, 43.3, 39.9, 37.1, 32.5, 32.1, 30.5, 28.6, 28.5, 27.7, 27.4, 26.5, 26.1, 23.4, 18.7, 17.6, 16.6, 16.4. HRMS (CH_3_OH) calcd for C_36_H_49_NNaO_6_^+^: *m*/*z* = 614.3412 ([M + Na]^+^), found: *m*/*z* = 614.3244 (Appendix A).

*p*-Tolyl 3β-hydroxy-11-oxo-olean-12-en-30-oate (**3e**): Yield 67.3%, ^1^H-NMR (500 MHz, DMSO-*d*_6_) (Appendix A): δ 7.23 (d, *J* = 8.5 Hz, 2H, Ar-H), 6.96 (d, *J* = 8.5 Hz, 2H, Ar-H), 5.42 (s, 1H), 4.30 (s, 1H), 3.01 (t, 1H, *J* = 6.0 Hz, 1H), 2.64–2.63 (m, 1H), 2.58–2.54 (m, 2H), 2.38–2.31 (m, 4H), 2.18–2.11 (m, 2H), 1.98–1.75 (m, 4H), 1.69–1.62 (m, 1H), 1.59–1.43 (m, 4H), 1.41–1.28 (m, 8H), 1.21–1.14 (m, 1H), 1.08–1.01 (m, 6H), 0.99–0.91 (m, 4H), 0.89–0.83 (m, 4H), 0.81–0.72 (m, 4H). ^13^C-NMR (125 MHz, DMSO-*d*_6_) (Appendix A): 199.5, 175.3, 169.6, 1 48.8, 135.4, 130.4 (2C), 127.9, 121.7 (2C), 77.0, 61.6, 54.5, 48.6, 45.3, 44.2, 43.3, 39.9, 37.8, 37.1, 32.5, 32.1, 30.7, 29.2, 28.7, 28.5, 27.8, 27.4, 26.5, 26.4, 26.2, 23.4, 20.8, 18.8, 18.5, 17.8, 17.6. HRMS (CH_3_OH) calcd for C_37_H_52_NaO_4_^+^: *m*/*z* = 583.3763 ([M + Na]^+^), found: *m*/*z* = 583.3676 (Appendix A).

4-Methoxyphenyl 3β-hydroxy-11-oxo-olean-12-en-30-oate (**3f**): Yield 63.6%, ^1^H-NMR (500 MHz, DMSO-*d*_6_) (Appendix A): δ 7.01–6.95 (m, 4H), 5.42 (s, 1H), 4.31 (s, 1H), 3.78–3.75 (m, 3H), 3.01 (t, 1H, *J* = 6.0 Hz, 1H), 2.58–2.54 (m, 1H), 2.36–2.34 (m, 1H), 2.18–2.12 (m, 2H), 1.97–1.72 (m, 4H), 1.68–1.62 (m, 1H), 1.57–1.44 (m, 4H), 1.43–1.25 (m, 12H), 1.19–1.14 (m, 1H), 1.08–1.02 (m, 9H), 1.01–0.92 (m, 3H), 0.91–0.82 (m, 2H), 0.80–0.71 (m, 2H). ^13^C-NMR (125 MHz, DMSO-*d*_6_) (Appendix A): 199.5, 175.5, 169.6, 157.3, 144.3, 127.8, 122.8 (2C), 115.0 (2C), 77.0, 61.6, 55.9, 54.5, 48.6, 45.3, 44.2, 43.3, 40.1, 42.5, 39.9, 37.8, 37.1, 32.5, 32.1, 30.7, 28.7, 28.5, 27.8, 27.4, 26.5, 26.2, 23.4, 18.8, 17.6, 16.6, 16.4. HRMS (CH_3_OH): calcd for C_37_H_52_NaO_5_^+^: *m*/*z* = 599.3712 ([M + Na]^+^), found: *m*/*z* = 599.3566 (Appendix A).

### 3.3. Purity Determination of the Compound

The purity of compound **3a** was analyzed on a COSMOSIL 5C_18_-MS-II column (250 mm × 10 mm) at 25 °C. We used H_2_O containing 0.1% trifluoroacetic acid (TFA) as mobile phase A and methanol containing 0.1% TFA as mobile phase B, with a flow rate of 3 mL/min. The elution program of compound **3a** was H_2_O (0.1% TFA): MeOH (0.1% TFA) = 5:95, and the detection wavelengths were set to 251 nm and 254 nm.

### 3.4. Cell Viability Assay

The effects of compounds on cells proliferation were evaluated using the 3-(4,5-dimethylthiazol-2-yl)-2,5-diphenyltetrazolium bromide (MTT) assay. It is based on the ability of succinate dehydrogenase in living cells to reduce MTT to an insoluble purple formamide precipitate, while dead cells do not [44]. The cells were placed in 96-well plates and treated overnight at 37 °C and 5% CO_2_. When the cells grew to about 80%, they were treated with different concentrations of compound **3a** for 48 h. Then, the culture medium was removed and MTT solution was added. After 4 h, the absorbance values were measured at 490 nm. In addition, the IC_50_ values were calculated with SPSS (Statistical Product and Service Solutions).

### 3.5. DAPI Studies Apoptotic Morphology

4′,6-diamidino-2′-phenylindole (DAPI) can enter cells through the cell membrane and dye the nucleus blue, which is used to detect the changes in the cell nuclear morphology. Normal nuclei are round and the chromatin is uniform, while the nuclei of apoptotic cells shrink and the cell outline is not clear [45]. HeLa cells were treated with compound **3a** for 24 h and fixed with paraformaldehyde for 15 min. The cells were stained with DAPI solution for half an hour and washed three times with PBS. Then the cells were photographed under ImageX-press R Micro XLS System (MD company, San Jose, CA, USA).

### 3.6. Apoptosis Was Detected Using Flow Cytometry

After incubating the cells in 6-well plates for one day, a concentration gradient of compound **3a** was added for 24 h. The cells were washed twice with PBS and then digested by adding trypsin. The supernatant was removed via centrifugation, and Annexin V-FITC buffer and Annexin-FITC solution were added. The percentage of the apoptotic cells was detected on a FACS Calibur flow cytometer (Beckman Dickinson & Co., Franklin Lakes, NJ, USA).

### 3.7. Wound Healing Migration Assay

HeLa cells (5.5 × 10^5^) were seeded in a 6-well plate and incubated for one day; they were then scratched and traced with the tip of a clean pipette. After washing the cells three times with PBS to remove the residue, a concentration gradient of compound **3a** was added for 24 h. Finally, after washing the cells twice with PBS, the migration of the cells was observed under an optical microscope and photographed.

### 3.8. Colony Formation Assay

HeLa cells (5.5 × 10^5^) were seeded in 6-well plates and placed in the incubator overnight, and the cells were treated with different concentrations of compound **3a** for 24 h. Then, the culture medium was replaced with fresh culture medium. The cells were grown for ten consecutive days; then, the culture medium was removed and the cells were washed twice with PBS. Finally, the cells were fixed with 4% paraformaldehyde solution for 30 min and stained with crystal violet (5%, *w*/*v*) for 30 min; the cells were observed under a light microscope and recorded.

### 3.9. Cell Cycle Arrest Assay

HeLa cells (5.5 × 10^5^) were spread in 6-well plates overnight; the cells were treated with different concentration of compound **3a** for 24 h. The cells were then washed twice with PBS and digested with trypsin. The supernatant was removed via centrifugation and fixed overnight in 70% ethanol. The cells were washed twice with PBS and resuspended in a 190 μL staining buffer containing 4 μL of 1 mg/mL PI (propidium iodide), 4 μL of 10 mg/mL RNaseA (ribonuclease), and 0.2 μL Tritonx-100. After staining for 20 min in the dark, the cell cycle distribution was detected using flow cytometry (Beckman Dickinson & Co., Franklin Lakes, NJ, USA) [46].

### 3.10. Determination of Intracellular Reactive Oxygen Species (ROS)

HeLa cells (5.5 × 10^5^) were spread in 12-well plates and incubated at 37 °C in 5% CO_2_ overnight. After treatment of HeLa with different concentrations of compound **3a** for 24 h, the culture medium in the wells was removed and the cells were washed twice with PBS. The cells were then stained with 20 μM 2′,7′-dichlorodihydrofluorescein diacetate (DCFH-DA) in the dark for 30 min. Finally, the cells were observed under the ImageXpress Mico XLS system (MD company, San Jose, CA, USA) and the intracellular reactive oxygen species content was determined under flow cytometry (Beckman Dickinson & Co., Franklin Lakes, NJ, USA).

### 3.11. Effects of Compound on Ca^2+^ Levels

HeLa cells (5.5 × 10^5^) were spread in a 6-well plate and incubated for 24 h; then, the cells were treated with different concentrations of compound **3a** for 24 h, and the cells were stained with the fluorescent probe Fluo-3 AM for 20 min and washed three times with PBS. Moreover, the nuclei were stained with DAPI in the dark for 30 min. Finally, the cells were observed under a fluorescence microscope (MD company, San Jose, CA, USA). Furthermore, the level of intracellular Ca^2+^ was quantified using flow cytometry (Beckman Dickinson & Co., Franklin Lakes, NJ, USA).

### 3.12. Effects on Mitochondrial Membrane Potential

Exponential growth phase HeLa cells were seeded into 12-well plates overnight. After the treatment of HeLa with different concentrations of compound **3a** for 24 h, the cells were then washed twice with PBS and stained with 200 μL of 5,5′,6,6′-Tetrachloro-1,1′,3,3′-tetraethyl-imidacarbocyanine iodide (JC-1) in the dark for 30 min; the cells were washed with PBS to remove residual dye solution. Then, the cells were photographed under a fluorescence microscope (MD company, San Jose, CA, USA) and the intensities of red and green fluorescence were measured with flow cytometry (Beckman Dickinson & Co., Franklin Lakes, NJ, USA).

### 3.13. Detection of Intracellular GSH Levels

HeLa cells (5.5 × 10^5^) were spread in 6-well plates, and intracellular GSH was detected using the GSH and GSSG assay kits (Beyotime, Biotechnology, China). The cells were digested down with trypsin after treatment with compound **3a**. Afterwards, the cells were lysed using the freeze-thaw method, and the lysate was centrifuged for 5 min to collect the supernatant. The absorbance was measured at 412 nm with a microplate reader, and then the GSH content was calculated.

### 3.14. Transcriptome Analysis

HeLa cells (5.5 × 10^5^) were spread in 6-well plates and placed in an incubator at 37 °C for 24 h. The cells were treated twice with IC_50_ concentration of compound **3a** for 24 h. The cells were then washed twice with pre-cooled PBS. An appropriate amount of RNA extract (Servicebio, China) was added, and the cells were fully lysed by blowing the liquid several times with a pipetting gun. Trizol reagent (Invitrogen Life Technologies, CA, USA) was then added to extract total RNA, and the concentration, quality and integrity of RNA were measured using NanoDrop spectrophotometer (Thermo Scientific, MA, USA). RNA sequence libraries were generated using the TruSeq RNA sample preparation kit (lllumina, San Diego, CA, USA). The library was also optimized using the AMpureXP system (Beckman Coulter, Beverly, CA, USA) to choose cDNA fragments of 200 bp in length. Moreover, the library fragments were quantified using Agilent high-sensitivity DNA analysis on a BioAnalyst 2100 system (Agilent, Santa Clara, CA, USA). Finally, the sequencing library was sequenced on the Hiseqplatform (lllumina) by Shanghai Personal Biotechnology Co., Ltd (Shanghai, China).

### 3.15. Western Blot Detection

After HeLa cells (5.5 × 10^5^) were seeded in 6-well plates and incubated for 24 h, the cells were treated with different concentrations of compound **3a**. Then, the cells were lysed with lysis buffer and centrifuged, and the protein concentration of each sample supernatant was measured with the BCA (bicinchoninic acid) assay. Load the sample protein in equal amount, and stop the sodium dodecyl sulfate-polyacrylamide gel electrophoresis after the bands are separated. The gel was transferred to poly (vinylidene difluoride) membranes (Millipore, Billerica, MA, USA) and blocked with 5% nonfat milk in TBST (20 mM Tris-HCl, 150 mM NaCl, 0.05% Tween 20, pH 8.0, Tween: polyoxyethylene monolaurate sorbaitan) buffer for 1 h. The polyvinylidene fluoride membranes were washed with TBST and incubated with the corresponding primary antibody overnight in a refrigerator at 4 °C. The secondary antibodies were then conjugated with horseradish peroxidase (1:1000 dilution) for 60 min at room temperature. Finally, the blots were visualized with the Amersham ECL (electrochemiluminescence) and western blotting detection reagents according to the manufacturer’s instructions.

### 3.16. Statistical Analysis

All data were expressed as the mean ± SD. Differences between two groups were analyzed using a two-tailed Student’s test. Differences with * *p* < 0.05 were considered statistically significant.

## 4. Conclusions

In this study, six GA analogs were synthesized and characterized using HRMS, ^1^H NMR and ^13^C NMR. In vitro cytotoxicity of compounds was carried out using the MTT assay, and the results of cloning experiments and scratch assay showed that compound **3a** resulted in an obvious inhibitory effect on the proliferation and migration of HeLa cells. The apoptotic assays confirmed that compound **3a** could induce apoptosis of HeLa cells. In the apoptosis mechanism diagram (Figure 11), Compound **3a** promotes an increase in intracellular Ca^2+^ levels and a decrease in GSH levels, leading to an increase in intracellular ROS content, which further resulted in impaired mitochondrial function, including a decrease in mitochondrial membrane potential and an increase in the protein ratio of Bax/Bcl-2. In addition, compound **3a** upregulated the expression of cell cycle arrest-related proteins p53 and p21, caused DNA damage, and activated the p53/p21 signaling pathway, leading to cell death by arresting in the S phase. Taken together, compound **3a** induced apoptosis in HeLa via the ROS-mediated mitochondrial dysfunction pathway. This work provides help for designing and synthesizing GA derivatives as potent anticancer candidate reagents.

## Data Availability

All data are available in the manuscript and Appendix A.

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
