# Peer review of "Evaluation of the Anticancer Activity and Mechanism Studies of Glycyrrhetic Acid Derivatives toward HeLa Cells"

_molecules, 2023, doi:10.3390/molecules28073164_

Round 1
Reviewer 1 Report
This is a complete and excellent study. The authors have covered all the routine anticancer experiments to evaluate their synthetic glycyrrhetinic acid derivatives. However, my suggestion and concern is why we should invest so much in synthesis part if we can get similar type of results with natural compounds. Other than this, this data also needs to be validated in a suitable in-vivo model in future studies.
Author Response
Reviewer 1#
This is a complete and excellent study. The authors have covered all the routine anticancer experiments to evaluate their synthetic glycyrrhetinic acid derivatives. However, my suggestion and concern is why we should invest so much in synthesis part if we can get similar type of results with natural compounds. Other than this, this data also needs to be validated in a suitable in-vivo model in future studies.
Response: because of the weak antitumor activity of glycyrrhetinic acid, there are many reports that structural modification of glycyrrhetinic acid can enhance biological activity through various synthesis. In future studies, we will investigate the antitumor activity of synthetic compounds in vivo model.
Reviewer 2 Report
The present paper "Evaluation of the anticancer activity and mechanism studies of glycyrrhetic acid derivatives toward HeLa cells" evaluates a series of glycyrrhetic acid derivatives as potential anticancer molecules. The authors describes the synthesis and the characterization of those compounds using HeLa cells as a model. The paper is interesting but there are some comments to be addressed before publication:
- in the section 3.2 the authors presented "In vitro cytotoxicity assay" of the synthesized compounds and they showed the results obtained from MTT assay. First of all, I would recommend to re-consider this paragraph because if MTT assay is performed, then the information you obtain is related to cell viability; if you want to get any information on cell cytotoxicity, you need to perform LDH assay then. Furthermore, in the Table 1, the unit should be indicated.
- Figure 3d: the authors reported "Gray value of FAK". Once a WB has been imaged, then it should be analyzed using densitometry to measure the relative amount of a specific protein for a given experimental sample on the blot and compare it with a control. The data should be presented after normalization.
- Figure 4c: western blot analysis should be presented after normalization.
- Figure 5a: Images are blurred. Please use images of good quality
- Figure 5c: the authors checked the expression levels of p38. Since they claim that in response to stress, p38 can be activated and phosphorylated, the phosphorylation of p38 should be evaluated as well. For the same figure, please provide also graph for the analysis of WB after normalization.
- Figure 6a: please provide better images.
- Figure 10: Please provide quantitative analysis of protein expression of PARP, Capase-3, Caspase-7, Bcl-2 and Bax, GPX4 after normalization.
Author Response
Reviewer 2#
The present paper "Evaluation of the anticancer activity and mechanism studies of glycyrrhetic acid derivatives toward HeLa cells" evaluates a series of glycyrrhetic acid derivatives as potential anticancer molecules. The authors describe the synthesis and the characterization of those compounds using HeLa cells as a model. The paper is interesting but there are some comments to be addressed before publication:
- in the section 3.2 the authors presented "In vitro cytotoxicity assay" of the synthesized compounds and they showed the results obtained from MTT assay. First of all, I would recommend to re-consider this paragraph because if MTT assay is performed, then the information you obtain is related to cell viability; if you want to get any information on cell cytotoxicity, you need to perform LDH assay then. Furthermore, in the Table 1, the unit should be indicated.
Response: We have revised section 3.2 part and provided the unit in Table 1.
- Figure 3d: the authors reported "Gray value of FAK". Once a WB has been imaged, then it should be analyzed using densitometry to measure the relative amount of a specific protein for a given experimental sample on the blot and compare it with a control. The data should be presented after normalization.
Response: The west blot was analyzed using densitometry in figure 3d after normalization.
- Figure 4c: western blot analysis should be presented after normalization after normalization.
Response: The west blot was analyzed using densitometry in figure 4c after normalization.
- Figure 5a: Images are blurred. Please use images of good quality
Response: We have provided the clearer images in figure 5a.
- Figure 5c: the authors checked the expression levels of p38. Since they claim that in response to stress, p38 can be activated and phosphorylated, the phosphorylation of p38 should be evaluated as well. For the same figure, please provide also graph for the analysis of WB after normalization.
I am sorry, in this experiment, we used p38MAPK, not p38, in the manuscript, change all p38 into p38MAPK. WB for Figure 5c has been revised after normalization.
- Figure 6a: please provide better images.
Response: We have changed the images in figure 6a.
- Figure 10: Please provide quantitative analysis of protein expression of PARP, Capase-3, Caspase-7, Bcl-2 and Bax, GPX4 after normalization.
Response: We have used densitometry to quantitative analysis of expression of PARP, Capase-3, Caspase-7, Bcl-2 and Bax, GPX4 in fugure 10.